# Reusing Waste Coffee Grounds in the Preparation of Porous Alumina Ceramics

Mihone Kerolli Mustafa [1],*, Ivana Gabelica [2], Vilko Mandić [3], Rea Veseli [2] and Lidija Ćurković [2],*

[1]  International Business College Mitrovica, 40000 Mitrovica, Kosovo
[2]  Faculty of Mechanical Engineering and Naval Architecture, University of Zagreb, Ivana Lučića 5, 10000 Zagreb, Croatia
[3]  Faculty of Chemical Engineering and Technology, University of Zagreb, Marulićev Trg 20, 10000 Zagreb, Croatia
*  Correspondence: m.kerolli@ibcmitrovica.eu (M.K.M.); lidija.curkovic@fsb.hr (L.Ć.); Tel.: +383-4-4235-479 (M.K.M.); +385-1-6168-183 (L.Ć.)

**Abstract:** Porous ceramics can be used in various industrial applications, such as thermal insulation, orthopedic implants, high-temperature filtration, lightweight structural components, and catalyst supports, etc., and can be obtained using various methods. In this study, the sacrificial fugitive method was used to prepare a porous alumina ceramic. The appropriate amount of sacrificial fugitive was combined with raw ceramic powder as a pore-forming agent, and was then evaporated or burned out either before or during the sintering process to create the desired pores. Various materials can be used as pore-forming agents; in this work, eco-friendly waste coffee grounds (WCG) were utilized. First, alumina ceramic green bodies were prepared via slip casting of 60 wt. % alumina suspensions with five different amounts of WCG (0 wt. %, 1 wt. %, 5 wt. %, 10 wt. % and 15 wt. %) and the dispersant Dolapix (0.2 wt. %), and using PVA (0.5 wt. %) as a binder for all solutions. The effect of the various amounts of WCG on the alumina ceramic green bodies, and subsequently on the obtained sintered ceramics, was tracked and validated through different analyses. Suspension viscosity was determined through a rotational viscometer. Simultaneous differential thermal and thermogravimetric (DTA/TGA) analyses were used to observe the thermal decomposition of WCG and to determine the sintering regime. After sintering, the density, porosity, and shrinkage of the samples were examined and calculated. In addition, the phase composition and crystallite size of all sintered samples were determined by powder X-ray diffraction (PXRD) analysis, as well as their morphology and composition using Scanning Electron Microscopy (SEM) equipped with energy dispersive X-ray spectroscopy (EDS). The results show that density decreased from 3.743 to 2.172 g/cm$^3$ and porosity increased from 6.12% to 45.52%, both with the increasing amount of WCG (from 0 wt. % to 15 wt. %).

**Keywords:** porous ceramics; alumina; waste coffee

## 1. Introduction

Wide and various applications of porous alumina ceramics, such as filters and membranes [1], catalyst supports [2], refractory materials [3], construction materials [4], thermal insulators [5], and many others, can be attributed to the fact that porous alumina can be formed into bodies with high porosity, high surface area, low density, high thermal and chemical resistance, as well as corrosion resistance. There are several manufacturing methods used to obtain porous ceramics, some of which are precipitation [6], replica technique [2], sacrificial template technique [7], a pore-forming agent method [8–10], uniaxial pressing [4], isostatic pressing [8], and gel-casting [11].

The supply of non-renewable resources is depleting more than ever because energy demands continue to rise. One possible solution to slowing down their consumption is by using waste materials in the production of new and improved materials. All areas

of society generate waste, from large industries to small households. Studies in various industry fields emerged in recent years, where waste was used to form materials with specific characteristics [12–18]. For example, Faridmehr et al. used fly ash, palm oil fly ash, waste ceramic powder, and granulated blast-furnace slag for alkali-activated mortars with promising environmentally friendly features, and appropriate strength and durability [19]. Khitab et al. found that about 27% of clay can be replaced with ceramic waste powder and waste brick powder in clay bricks; this can preserve a large quantity of natural clay without compromising the quality of the bricks [20]. In another study, by Collivignarelli et al., construction and demolition waste, residues from waste treatment, metallurgical industry by-products, and others, were used as substitutes for either fine or coarse aggregates in concrete [18]. Naturally, part of that waste can be used as pore-forming agents in the preparation of porous ceramics. Studies show that there is huge potential for absorbing waste and saving raw materials when using waste material to produce ceramics: some of the most prominent waste materials are inorganic—fly ash, various sludges and slags, and mud, while others are organic—rice husk, rice-husk ash, eggshells and coffee grounds [11,21,22]. Torman et al. produced lightweight anorthite ceramics utilizing admixtures of eggshell waste, fireclay, and expanded polystyrene, and showed these ceramics could be suitable for use in various applications requiring elevated temperatures [17]. Cao et al., used waste fly ash and natural bauxite, with $AlF_3$ and $V_2O_5$ as additives, to obtain a mullite ceramic membrane support with increased open porosity and without strength degradation [23]. Ali et al. [9] used graphite waste from batteries combined with a fugitive material technique to produce porous alumina ceramics that had increased porosity and deformation tolerance; however, they had less mechanical strength, resulting in a higher strain at break. In another study by the same authors, commercial rice-husk ash and a solid-state technique were used to obtain a porous ceramic composite [12]. With higher ratios of rice-husk ash, the composite's mechanical properties were improved. Eliche-Quesada et al. achieved improved thermal insulation properties in ceramic bricks utilizing various industrial wastes [13]. Paper waste, sawdust, and corn starch were used to manufacture low-cost porous ceramics by Salman et al. [10], further confirming that the most suitable pore-forming agent is paper waste, which provided good mechanical properties, porosity, and permeability of the obtained porous ceramic composite. In the case of unprocessed plant-based waste, which is even more environmentally friendly, Liu et al. [8] used walnut-shell powder as a pore-forming agent together with alumina sol impregnation to obtain porous alumina ceramics with improved crushing strength and thermal insulation performance. Differently, Alzukaimi et al. used sunflower-seed shells as the fugitive template material for macroporous alumina ceramics through the space holder technique, obtaining a wide range of porosity and corresponding high mechanical strengths, also leading to a wide range of potential applications [22].

One of the most popular and often consumed beverages in the world is coffee, and subsequently, huge amounts of coffee waste are generated in the brewing process. This waste can also be used as a pore-forming agent in porous ceramic production [14]. Sena da Fonseca et al. [4] studied the effect of coffee waste in structural construction ceramics, concluding that it can be used as a secondary clay raw material that improves the thermal insulation properties of bricks. Alzukaimi et al. [7] prepared macroporous alumina ceramics with coffee waste and obtained high open porosity of the ceramic material along with good mechanical strength. Furthermore, spent coffee grounds were utilized as a filler material in lightweight clay ceramic aggregates by Andreola et al. [15]. The resulting material showcased interesting properties that make it applicable for both urban and agricultural purposes.

In this paper, waste coffee grounds and slip casting methods were used to produce porous alumina ceramics [16]. The present paper aims to provide further knowledge on the influence of varying amounts of waste coffee grounds used to produce alumina ceramics with varying porosities. The preceramic suspensions with different amounts of waste coffee grounds were described by rheological measurements, while the sintering regime was

determined after density experiments and DTA-TGA analysis of the thermal decomposition coffee grounds.

## 2. Materials and Methods

### 2.1. Suspension Preparation

Alumina suspensions were prepared with 60 wt. % of alumina powder. High-purity $Al_2O_3$ powder was used, with an average particle size of 300–400 nm (Alcan Chemicals, Green, OH, USA). Dolapix CE64 (Zschimmer and Schwarz Chemie GmbH, Lahnstein, Germany), the ammonium salt of poly (methacrylic) acid (polyelectrolyte), was used as a dispersant to stabilize highly concentrated alumina suspensions. Polyvinyl alcohol, PVA (Sigma Aldrich, St. Louis, MO, USA), was added to the suspension as a binder. Different amounts (0, 1, 5, 10 and 15 wt. %) of waste coffee grounds (WCG), obtained from a household coffee machine with a grinder, were added to alumina suspensions as the pore-forming agent. The sample composition of the various prepared $Al_2O_3$ suspensions is shown in Table 1.

**Table 1.** The composition of the prepared $Al_2O_3$ suspensions in mass fractions (wt. %).

| wt. % | | | | | |
|---|---|---|---|---|---|
| $Al_2O_3$ + WCG | $H_2O$ | $Al_2O_3$, in the Powder Mixture | WCG, in the Powder Mixture | DOLAPIX CE64 | PVA |
| 60 | 40 | 100 | 0 | 0.2 | 0.5 |
| 60 | 40 | 99 | 1 | 0.2 | 0.5 |
| 60 | 40 | 95 | 5 | 0.2 | 0.5 |
| 60 | 40 | 90 | 10 | 0.2 | 0.5 |
| 60 | 40 | 85 | 15 | 0.2 | 0.5 |

The process of homogenization started by adding 0.2 wt. % DOLAPIX CE 64 to distilled water; after this dissolved in the water, the ceramic $Al_2O_3$ powder and 0.5 wt. % PVA were added while being stirred with a glass rod. Different amounts of waste coffee grounds were added to certain suspensions. Following this, the suspensions were poured into a planetary ball mill (PM 100, Retsch GmbH, Haan, Germany) container alongside ten ceramic balls for better mixing and homogenization. (These balls, as well as the container, are made of aluminum oxide ceramics to avoid contamination of suspensions). The suspensions were homogenized at a speed of 300 revolutions per minute for 90 min. After homogenization, the ceramic balls were separated from the suspensions by filtering and the air bubbles were removed in an ultrasonic bath (BRANSONIC 220, Branson Ultrasonics Corp., Brookfield, CT, USA) for 15 min, to ensure the rheological measurements were as reliable as possible.

### 2.2. Determination of Rheological Properties

The rheological properties of the prepared ceramic suspensions were determined using a rotational viscometer, ViscoQC$^{TM}$300 (Anton Paar, Graz, Austria) in a measuring cup, C-CC12, with a measuring bob, B-CC12. The shear rate was gradually increased from 0.1 to 180 s$^{-1}$, and then reduced back to 0.1 s$^{-1}$ divided into 50 equal time frames. The rheological parameters were recorded just before each shear rate change.

### 2.3. Sintering of Green Bodies

In detachable plaster molds, green bodies were formed by casting the prepared $Al_2O_3$/WCG suspensions. Gypsum absorbs water easily and quickly, which is why it is often used to make molds. The cast samples were then dried in these plaster molds at room temperature for 48 h. After removing them from the molds, the green bodies were dried for another 3 h in the Instrumentaria ST05 dryer (Instrumentaria d.d., Sesvete, Croatia) at 100 °C.

The obtained green bodies were sintered in an electric furnace (Nabertherm P310, Bremen, Germany) according to the course of thermal decomposition of waste coffee grounds, which was determined by Differential Thermal Analysis (DTA) and Thermo-Gravimetric Analysis (TGA) using a simultaneous DTA/TGA device STA409 (Netzsch, Selb, Germany).

*2.4. Characterization of Sintered Alumina Ceramics*

The phase composition of $Al_2O_3$ powder raw material, sintered alumina ceramic samples with different amounts of waste coffee grounds (WCG) (0 wt. %, 1 wt. %, 5 wt. %, 10 wt. % and 15 wt. %), WCG raw, and WCG thermally treated at 1600 °C, were determined by powder X-ray diffraction, PXRD (Shimadzu XRD6000, Shimadzu Corporation, Japan) with CuK$\alpha$ radiation. A step size of 0.02 degrees between 10° and 80°, 2θ, and a counting time of 0.6 s were used, under an accelerating voltage of 40 kV and a current of 30 mA.

The bulk density of the sintered alumina samples was determined by the Archimedes method (Mettler Toledo GmbH, Switzerland, density kit MS-DNY-43) according to ASTM C373-88. The density of the $Al_2O_3$/WCG samples was calculated using the following equation:

$$\rho = \frac{m}{V} \tag{1}$$

where $m$ (g) is the mass of the sintered sample, and $V$ ($cm^3$) is the volume of the sintered sample.

The relative density of the sintered samples was calculated by the following equation:

$$\rho_{relative} = \frac{\rho_{Archimeds}}{\rho_{theoretical}} \cdot 100 \% \tag{2}$$

while the relative porosity was calculated as a difference between 100% density and relative density (%).

In addition to density ($\rho$, g cm$^{-3}$), relative density (*R. D.*, %), porosity (*P*, %) and shrinkage (*S*, %) values were also calculated with standard deviation values. Relative density and porosity were determined according to the following expression:

$$R.D., \% = \left( \frac{\rho_{Arhimedova}}{\rho_{teorijska}} \right) \cdot 100 \tag{3}$$

The theoretical density of aluminum oxide is 3.987 g cm$^{-3}$ [24].

The porosity (*P*, %) of the sintered alumina samples was determined according to the following equation:

$$P, \% = (1 - R.D.) \cdot 100 \tag{4}$$

where *P* (%) is the porosity and *R. D.* (%) is the relative porosity.

The shrinkage (*S*, %), i.e., the decrease in dimensions after sintering, is expressed in percentages, and was determined by measuring the change in the dimensions of the samples before and after sintering, according to the expression:

$$S, \% = \frac{d_s}{d - d_s} \cdot 100 \tag{5}$$

where *S* (%) is the shrinkage, *d* (mm) is the length of the sintered sample, and $d_s$ is the length of the corresponding green body.

The morphology of the prepared sintered samples was determined according to a standard ceramographic technique [21] using a scanning electron microscope (SEM), and the elemental composition of the sintered alumina ceramic samples was studied by scanning electron microscopy with an energy dispersive X-ray spectroscopy (EDS) (Tescan Vega TS5136LS, Brno, Czech Republic).

## 3. Results and Discussion

### 3.1. Rheological Measurements

Rheological properties were examined using rheological flow curves which show the dependence of shear rate ($\gamma$) on viscosity ($\eta$). These are used to predict the nature of interactions between particles in the suspension. Flow curves for alumina suspension without the addition of waste coffee grounds (WCG), as well as alumina suspensions with the addition of 1 and 5 wt. % of WCG, are shown in Figure 1.

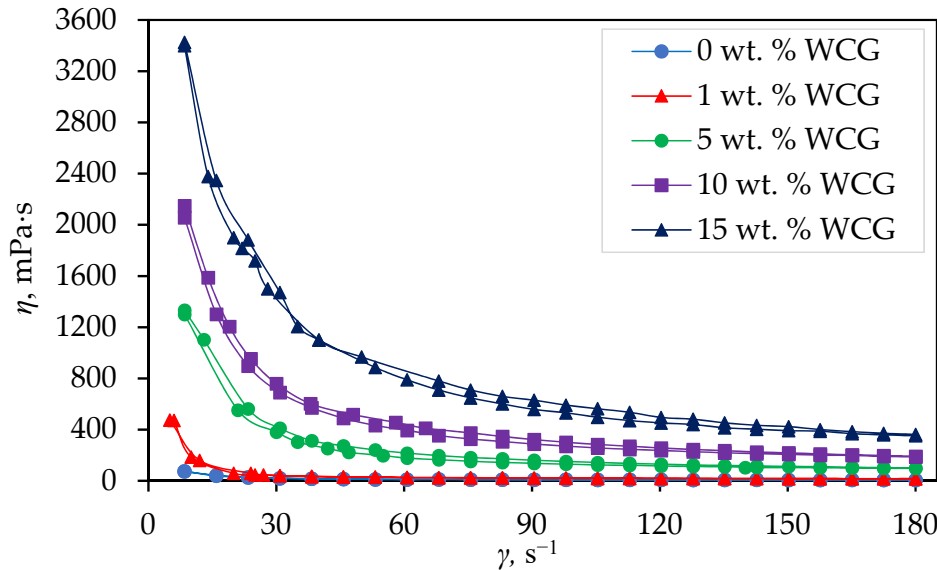

**Figure 1.** Effect of amount of waste coffee grounds (WCG) on rheological flow curves of the prepared alumina ceramic suspensions.

The obtained results show that the suspension viscosity decreases with an increase in the shear rate, that is, all suspensions show typical pseudoplastic behavior, characteristic of non-Newtonian fluids. Suspension stability was estimated by the viscosity measurements at two shear rates, 50 and 100 s$^{-1}$. The shear rate of 50 s$^{-1}$ is the exact shear rate usually achieved during gravity slip casting, while the shear rate of 100 s$^{-1}$ was chosen for the purpose of a better comparison. The obtained viscosity values of prepared alumina suspensions with the addition of waste coffee grounds (WCG) are shown in Table 2. The viscosity of prepared suspensions increased with the addition of waste coffee grounds.

**Table 2.** Measured viscosity for the prepared Al$_2$O$_3$/WCG ceramic suspensions.

| wt. (WCG in Powder Mixture), % | $\eta$ (mPa·s) | |
|:---:|:---:|:---:|
| | $\gamma$, 50 s$^{-1}$ | $\gamma$, 100 s$^{-1}$ |
| 0 | 11.8 | 8.9 |
| 1 | 29.4 | 19.2 |
| 5 | 241.4 | 143.4 |
| 10 | 518.1 | 283.9 |
| 15 | 967.6 | 563.4 |

The diagram in Figure 1 shows the apparent viscosity at the shear rate of the app. 50 s$^{-1}$, which is the shear rate of gravity slip casting.

### 3.2. Differential Thermal Analysis (DTA) and Thermo-Gravimetric Analysis (TGA)

DTA/TGA analyses (Figure 2) of waste coffee grounds revealed a single endothermic event up to 100 °C, related to the evaporation of the moisture and accompanied by ~0.5% of mass loss. Following this, up to 200 °C, there was another single endothermic event related

to the release of chemisorbed moisture, accompanied by ~0.25% of mass loss. The DTA shows that, above 200 °C to about 450 °C, there was a single broad exothermic event related to the burning of the organic material, accompanied by TGA mass loss in what appears to be a three-step process: the first, from 200 °C to about 320 °C, the second from 320 °C to about 390 °C, and the third from 390 °C to about 450 °C. From 400 °C to about 750 °C, the DTA signal is dominated by a broad endothermic envelope attributed to the subsequent release of the decomposition residuals. At 800 °C, ambiguous overlapped endothermic and exothermic effects of small intensity are observed without mass loss. Above 900 °C, carbonaceous residuals burn out and gas out, which slightly increases mass loss.

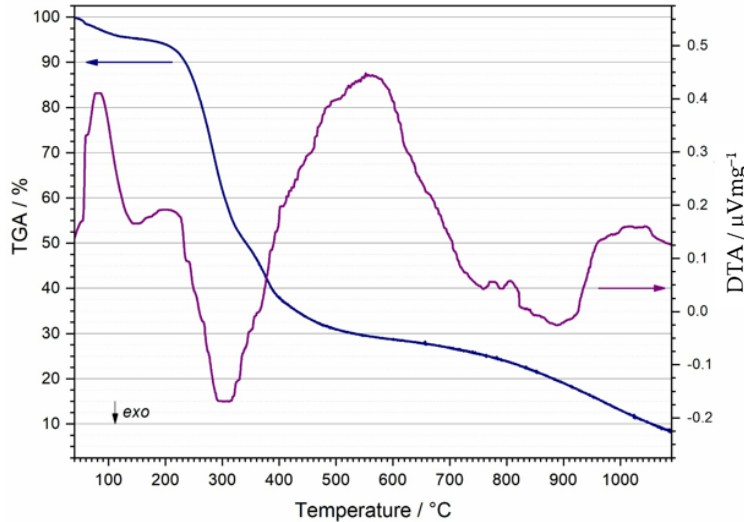

**Figure 2.** Differential Thermal Analysis (DTA) and Thermo-Gravimetric Analysis (TGA) curves of the waste coffee grounds.

According to the DTA/TGA curves, the sintering process is defined, as shown in Figure 3.

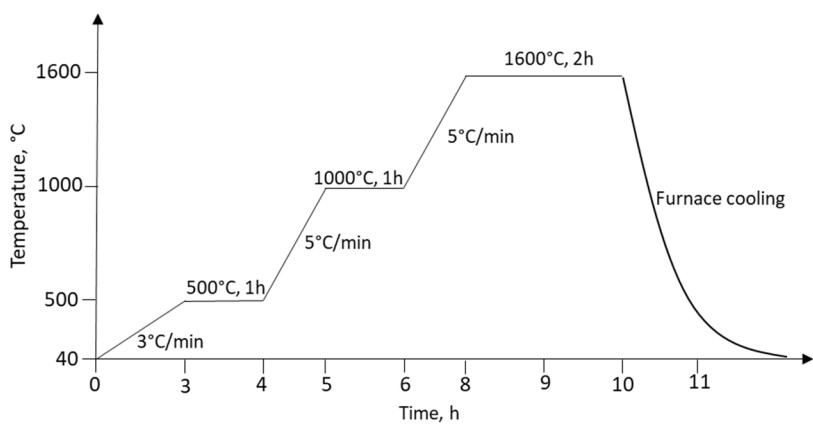

**Figure 3.** Scheme of the sintering regime of $Al_2O_3$/WCG ceramics.

First, heating to 500 °C with a heating rate of 3 °C·min$^{-1}$ and 1 h holding time at 500 °C, which enabled the organic material to burn out. Next, the samples were heated to 1000 °C with a heating rate of 5 °C·min$^{-1}$ and 1 h holding time at 1000 °C, allowing carbonaceous residuals to burn out and gas out. Following this, the samples were heated to a sintering temperature of 1600 °C at a heating rate of 5 °C·min$^{-1}$ and 2 h holding time at 1600 °C. After the cooling of the furnace, the sintered samples were removed for further characterization.

The results for the PXRD of $Al_2O_3$ powder raw and alumina ceramic samples with different amounts of waste coffee grounds (WCG) (0 wt. %, 1 wt. %, 5 wt., 10 wt. and 15 wt. %), WCG raw, and WCG thermally treated at 1600 °C, are shown in Figure 4. The XRD analysis shows that raw WCG is amorphous. The XRD for thermally treated coffee shows Mg-spinel (ICDD PDF#21-1152) and magnesium oxide periclase (ICDD PDF#45-0946) crystallize (Figure 4). These are high-temperature phases that can withstand the high-temperature thermal treatment and are, therefore, concentrated as organic phases are burned out of the sample. A minor calcium phosphate phase (ICDD PDF#09-0348) is visible in the XRD (Figure 4, marked yellow).

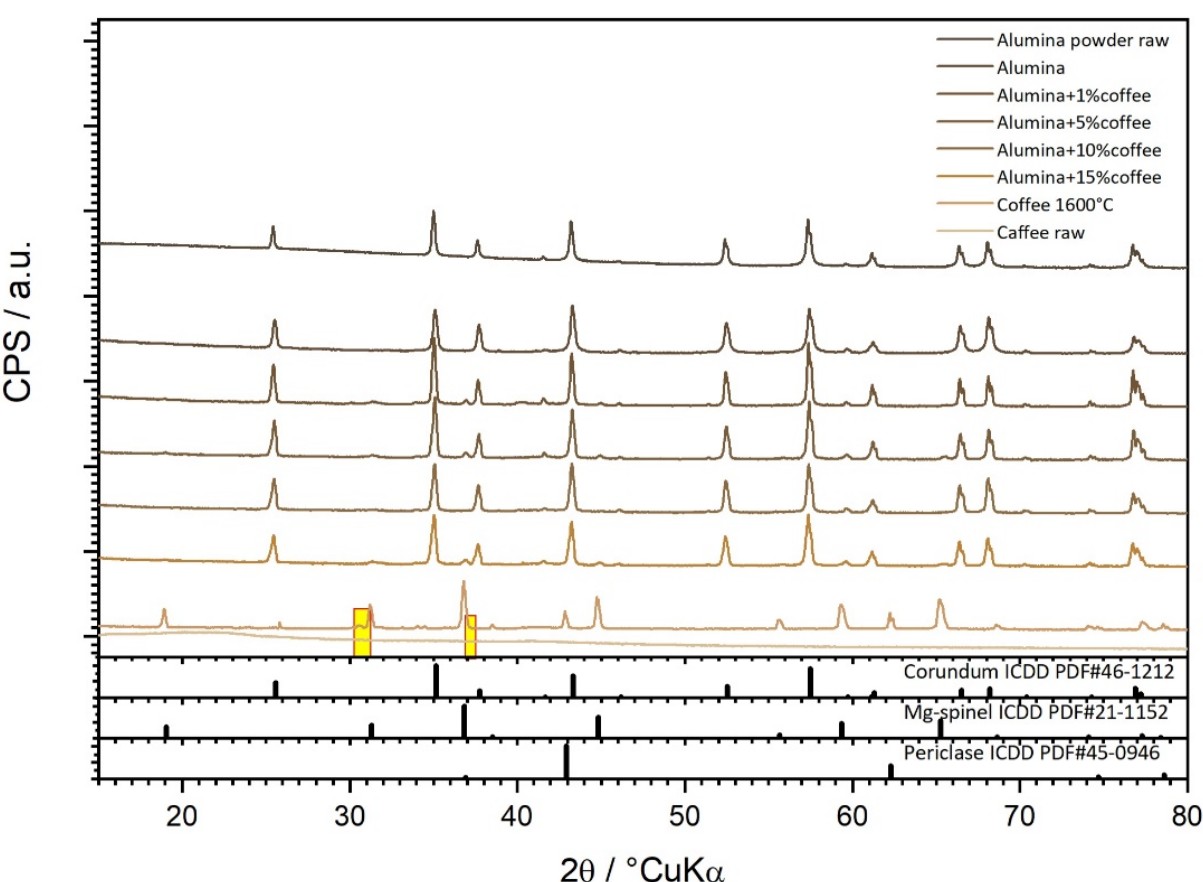

**Figure 4.** XRD patterns of $Al_2O_3$ powder raw, alumina ceramic samples with different amounts of WCG (0 wt. %, 1 wt. %, 5 wt., 10 wt. and 15 wt. %), WCG raw, WCG thermally treated at 1600 °C.

The crystallite size of all sintered alumina samples was measured from XRD patterns (Figure 4) using Scherrer's method. Generally, for crystallites up to 100 nm in size, the application of the Scherrer equation yields very reliable and precise estimations of crystallite sizes. However, this does not limit its broader range of applicability, where in the range 20–200 nm, the precision is still high. Moreover, for trends where precision is less important, the range can even be broadened up to 300 nm. In any case, the Scherrer equation suffers from a set of other assumptions, so stating that the values are correct up to 100 nm—but not above this value—is simply inappropriate. In terms of trend, and similar to authors of numerous publications [25], our aim was to demonstrate this. The obtained results are presented in Table 3. Alumina crystallite sizes (Table 3) increase with the addition of 1 wt. % of WCG. By further increasing the amount of WCG (5, 10 and 15 wt. %), the alumina crystallite size is not in linear correlation.

**Table 3.** Value of crystallite sizes of all sintered alumina samples determined from XRD patterns.

| wt. (Waste Coffee Grounds in Alumina), % | 0 | 1 | 5 | 10 | 15 |
|---|---|---|---|---|---|
| Crystallite Size, nm | 135 | 198 | 158 | 148 | 164 |

Figure 5A,B shows the values of obtained density ($\rho$, g cm$^{-3}$), relative density (*R. D.*, %) and total porosity (*P*, %). Figure 5C shows the results of shrinkage (*S*, %) of the various Al$_2$O$_3$/WCG ceramics.

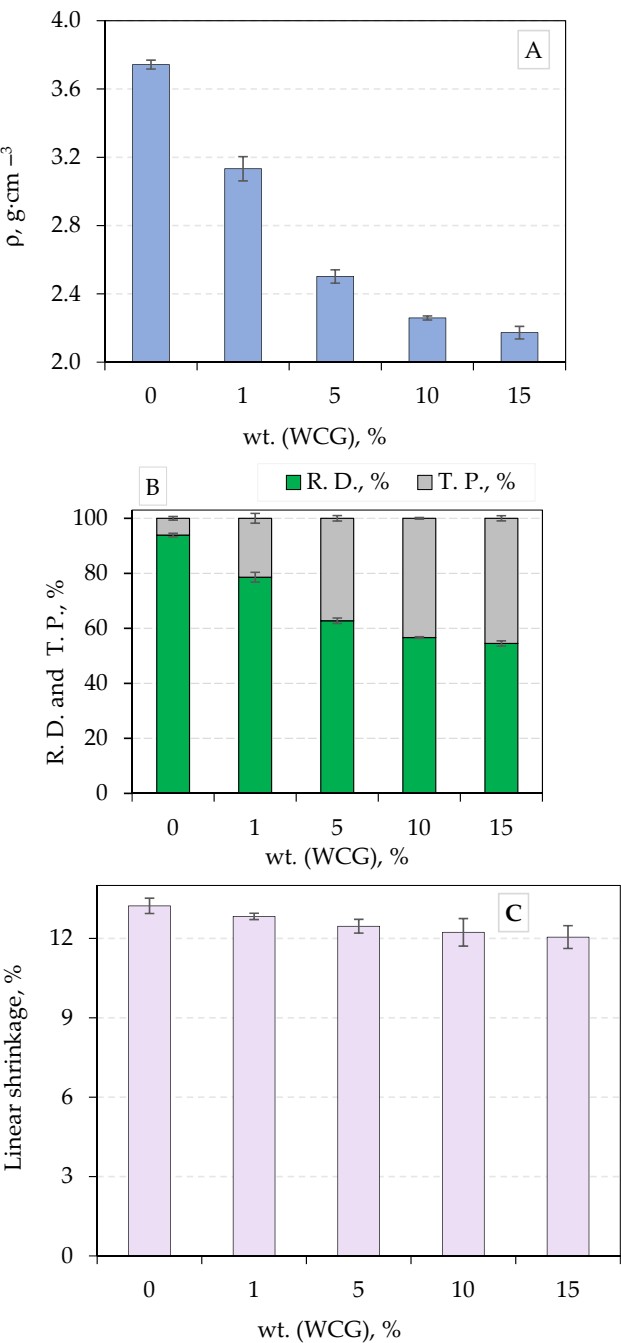

**Figure 5.** Effect of amount of waste coffee grounds (WCG) on the following properties of sintered alumina ceramics: (**A**) density ($\rho$, g·cm$^{-3}$), (**B**) relative density (R. D., %), total porosity (T. P., %) and (**C**) linear shrinkage (LS, %).

From the obtained results shown in Figure 5A, a decrease in the density of sintered samples of $Al_2O_3$ ceramics from 3.743 to 2.172 g cm$^{-3}$ can be observed with the increase in the WCG amount from 0 to 15 wt. %. The same trend is visible for relative density (Figure 5B). A significant increase in the porosity of the sintered samples of the $Al_2O_3$ ceramics can also be observed, from 5.72 to 45.29% (Figure 5B) with the increase in WCG amount from 0 to 15 wt. %. Therefore, it can be assumed that the pores are formed by the combustion of WCG during sintering. The results of the shrinkage of the $Al_2O_3$ ceramic samples (Figure 5C) show a small change with the addition of WCG from 0 wt. % to 20 wt. %. Greater dispersion in the results is present only in the case of relative density and porosity, especially in samples with 1 and 15 wt. % of WCG. Sena et al. (2014) reported that an increasing amount of coffee waste in brick manufacture resulted in a worsened workability state [4,26], while mixtures with 10 wt.% and 15 wt.% had an adequate workability and showed desirable characteristics for practical use.

The microstructure of the samples was analyzed with a scanning electron microscope (SEM). Figure 6A–E show images of fracture surfaces of the various $Al_2O_3$/WCG ceramic samples, WCG raw (Figure 6F), and WCG thermally treated at 1600 °C (Figure 6G).

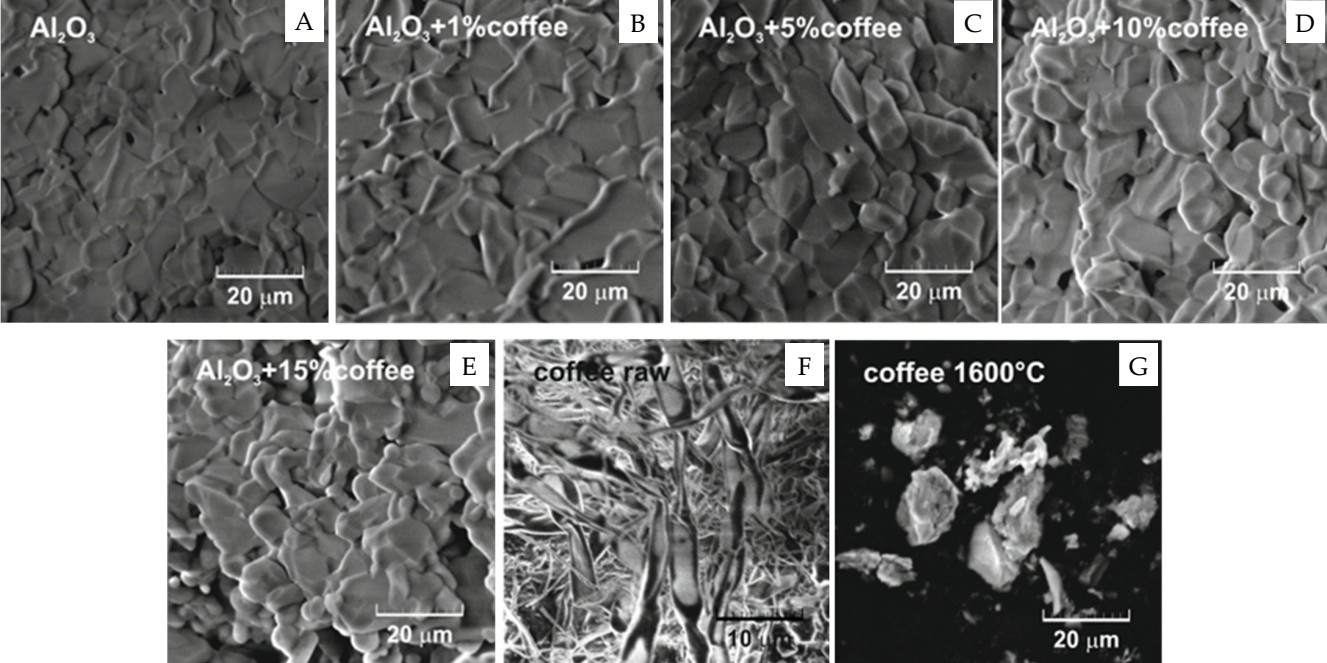

**Figure 6.** SEM micrographs of fracture surface morphology of $Al_2O_3$ ceramics with different waste coffee ground (WCG) content: (**A**) 0 wt. %, (**B**) 1 wt. %, (**C**) 5 wt. %, (**D**) 10 wt. %, and (**E**) 15 wt. %, prepared by slip casting and sintered at 1600 °C; and (**F**) WCG raw and (**G**) WCG thermally treated at 1600 °C.

By analyzing the obtained microstructure data shown in Figure 6A–E, it can be observed that the microstructure porosity increases with the increase in the amount of WCG. This is to be expected, considering that WCG is used as a pore-forming agent. From the morphology point of view, SEM micrographs show the grainy appearance and relatively dense packing, typical of alumina bodies (Figure 6). With the increase in coffee content, the only element that appears to change is the grain size distribution. The addition of coffee changes the overall uniformity and monomodal size distribution of pure alumina to a broader distribution. The grains seem to be of a similar shape. Such a phenomenon is in line with the burn out of the organic phase. For the case of WCG (Figure 6F), SEM shows networked amorphous matter, while for the case of thermally treated WCG (Figure 6G), the inorganic constituents of WCG remain, forming clearly formed irregular agglomerated particles with a crystalline appearance.

Figure 7 presents EDS spectra of Al, O, K, and Mg of $Al_2O_3$ powder raw, alumina ceramic samples with different amounts of WCG (0 wt. %, 1 wt. %, 5 wt., 10 wt. and 15 wt. %), WCG raw, and WCG thermally treated at 1600 °C.

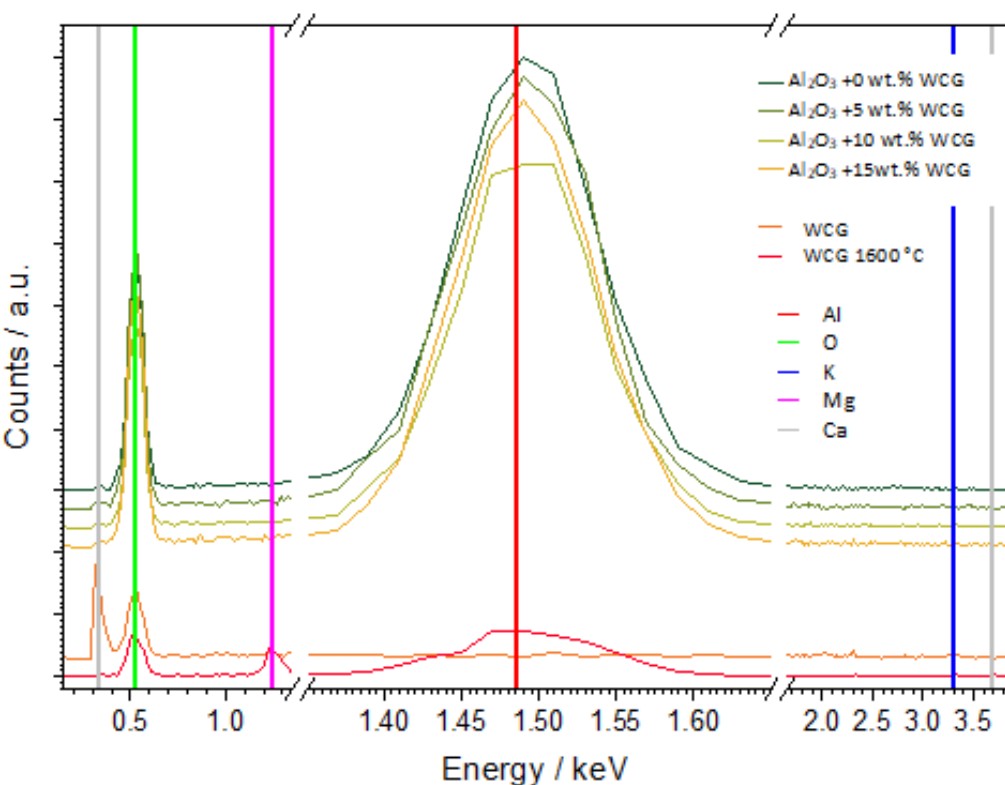

**Figure 7.** EDS spectra of Al, O, K, and Mg of $Al_2O_3$ powder raw, alumina ceramic samples with different amounts of WCG (0 wt. %, 1 wt. %, 5 wt., 10 wt. and 15 wt. %), WCG raw, and WCG thermally treated at 1600 °C.

Generally, the WCG composition resembles numerous organic constituents and few inorganic elements, such as Ca, K, and Mg. We observed, by EDS, that raw dried WCG powder is rich in Ca (Figure 7). For the case of thermally treated WCG, EDS shows the presence of Mg. The "changing" of the EDS coffee composition before and post thermal treatment makes much more sense when compared with the XRD results of coffee before and post thermal treatment (Figure 4). Specifically, XRD shows raw WCG as amorphous (Figure 4), which is in line with EDS; however, EDS resolution allows only the observation of Ca traces in a majorly organic matrix. For the case of thermally treated WCG, XRD shows that Mg-spinel (ICDD PDF#21-1152) and magnesium oxide periclase (ICDD PDF#45-0946) crystallize (Figure 4). These are high-temperature phases that can withstand the high-temperature thermal treatment and are, therefore, concentrated as organic phases burn out of the sample. This is in line with EDS, where only Mg remains visible (Figure 7). Ca is not visible in the EDS of thermally treated coffee, whereas a minor calcium phosphate phase (ICDD PDF#09-0348) is visible in the XRD (Figure 4). In this case, the calcium phosphate grains may not have been noticed or collected during EDS analysis at higher magnification, due to the distribution of the calcium phosphate grains.

Initial raw alumina powder only shows corundum (Figure 4). In the case of the alumina sintered samples, alumina without the addition of WCG does not show any phases other than corundum (Figure 4). Bearing in mind that thermally treated WCG contains Mg-spinel, the addition of different contents of WCG in the alumina, and subsequent thermal treatment, should reflect an increase in Mg-spinel; Mg provided from the Mg-spinel does not react with alumina. From the fact that the increase in Mg-spinel is not linear to WCG content, it can be concluded that Mg can be doped into alumina with the presence

of Mg-spinel. The solubility level of Mg in the alumina depends on its diffusion ability. The diffusion could be more efficient for the case of smaller alumina powder and WCG grains and their denser packing; i.e., the smaller presence of Mg-spinel in sintered bodies (at greater WCG contents) is an indicator of the quality of the grinding, processing, and shaping. These chemical changes in the alumina bodies with different WCG contents are not visible from the EDS spectra; rather, EDS points out the existence of Al and O only for all samples. Crystallite sizes suggest alumina grains grow upon the addition of 1 wt. % of WCG, while the results show that the grain growth mechanism is not linear when the amount of WCG increases by more than 1 wt. %.

## 4. Conclusions

In this work, porous $Al_2O_3$ ceramics were prepared with the addition of 0, 1, 5, 10 and 15 wt. % waste coffee grounds. Green bodies were formed through the slip casting method, followed by sintering in an electric furnace at a temperature of 1600 °C. The microstructure, phase, and chemical composition of the sintered samples were analyzed, and their density, relative density, porosity, and shrinkage were determined. From the obtained results, it can be concluded:

- All suspensions show a decrease in apparent viscosity with an increase in shear rate, which means that the suspensions belong to the group of non-Newtonian fluids, that is, the suspensions show pseudoplastic behavior.
- From the obtained diagrams of the dependence of apparent viscosity ($\eta$, mPa s) on shear rate ($\gamma$, s$^{-1}$), it is evident that samples with 0 wt. % waste coffee grounds at a shear rate of 50 s$^{-1}$ have viscosity values of 11.8 mPa s, while samples with 15 wt. % of waste coffee grounds at a shear rate of 50 s$^{-1}$ have values of around 967.6 mPa s, from which it can be concluded that increasing the waste coffee ground amount also increases the apparent viscosity.
- DTA/TGA analyses of waste coffee grounds revealed multistep endothermic and exothermic events related to the evaporation and release of the moisture, burning of the organic material, the subsequent release of the decomposition residuals, and carbonaceous residuals burnouts accompanied by mass loss. According to DTA/TGA measurements, a suitable course of thermal treatment for sintering was implemented.
- It was found that increasing the waste coffee ground amount increases the porosity of sintered $Al_2O_3$ ceramics.
- The crystallite size of alumina grains grows upon the addition of 1 wt. % of WCG.
- By increasing the waste coffee ground amount, the density and relative density of $Al_2O_3$ ceramics decrease.
- The porosity of the $Al_2O_3$ ceramic samples increased significantly with the waste coffee ground amount, and it is assumed that these pores are formed by the combustion of waste coffee grounds during sintering.
- The shrinkage of the obtained alumina ceramic samples is barely affected by the amount of the waste coffee grounds.
- The mechanical and tribological properties of alumina ceramics with addition different amounts of WCG will be investigated in future research work.

**Author Contributions:** Conceptualization, L.Ć., M.K.M. and R.V.; methodology, M.K.M., L.Ć. and I.G.; validation, L.Ć., M.K.M. and I.G.; formal analysis, I.G., V.M. and L.Ć.; investigation, M.K.M., I.G. and L.Ć.; resources, L.Ć.; data curation, M.K.M., L.Ć. and V.M.; writing—original draft preparation, M.K.M., L.Ć., V.M., I.G. and R.V.; writing—review and editing, M.K.M., L.Ć. and V.M.; visualization, M.K.M. and L.Ć.; supervision, L.Ć.; project administration, L.Ć.; funding acquisition, L.Ć. All authors have read and agreed to the published version of the manuscript.

**Funding:** This work was fully supported by the Croatian Science Foundation within the project IP-2016-06-6000: Monolithic and Composite Advanced Ceramics for Wear and Corrosion Protection (WECOR).

**Institutional Review Board Statement:** Not applicable.

**Informed Consent Statement:** Not applicable.

**Data Availability Statement:** The data presented in this study are available upon request from the corresponding author.

**Conflicts of Interest:** The authors declare no conflict of interest.

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
