# Peer review of "Reusing Waste Coffee Grounds in the Preparation of Porous Alumina Ceramics"

_sustainability, doi:10.3390/su142114244_

Round 1
Reviewer 1 Report
The authors reported a porous alumina ceramic fabricated by the sacrificial fugitive method, and some results were obtained. However, mainly due to the lack of sufficient innovation, this manuscript in the present version was regarded to be not appropriate for publication. And some comments were presented as follows.
1. Compared with the results of previously reported literatures, this method has a complex process and high synthesis temperature. Moreover, the property characterization of the product (such as mechanical property, thermal conductivity and so on) is lacking in the present paper. So, I do not think the similar work is enough novel for publication.
2. The effect of the amount of Dolapix CE64 and PVA on the rheological properties of the slurry, and the microstructure and mechanical properties of the final product should be considered.
3. In Figure 5B, error bars should be added.
4. XRD pattern of Al2O3 ceramics should be added.
5. In Figure 6, the grain size distribution of Al2O3 ceramics with different waste coffee grounds should be provided.
6. In line 278-279, It was found that increasing the waste coffee grounds amount reduces the crystal grains of sintered Al2O3 ceramics and increases porosity. Please explain why?
7. In Figure 5C, the shrinkage of the obtained foam is almost not affected by the amount of the waste coffee grounds. This is not consistent with the conclusion (Line 285-286).
8. References should be formatted in accordance with ‘Instructions for Authors’.
9. In addition, the English expression needs to be further improved by a professional scientific writer.
Author Response
Dear Editor and Reviewer,
Thank you for the review. We have done our best to correct the article and fulfill all the requirements and questions this time. We are resubmitting the corrected manuscript as suggested by the reviewers. The added or changed text was highlighted in “yellow”.
Our responses are as follows:
Reviewer #1:
The authors reported a porous alumina ceramic fabricated by the sacrificial fugitive method, and some results were obtained. However, mainly due to the lack of sufficient innovation, this manuscript in the present version was regarded to be not appropriate for publication. And some comments were presented as follows.
- Compared with the results of previously reported literatures, this method has a complex process and high synthesis temperature. Moreover, the property characterization of the product (such as mechanical property, thermal conductivity and so on) is lacking in the present paper. So, I do not think the similar work is enough novel for publication.
Response - The present paper aimed to provide further knowledge on the influence of different waste coffee grounds amounts used to produce alumina ceramics with different porosity. Mechanical and other properties of alumina ceramics with the addition of different amounts of WCG will be investigated in future research work. In addition, valuable information of other alumina ceramics samples properties (density, porosity, shrinkage, phase composition, crystallite size, morphology as well as EDS analysis) are provided.
- The effect of the amount of Dolapix CE64 and PVA on the rheological properties of the slurry, and the microstructure and mechanical properties of the final product should be considered.
Response - Thank you for your advice. The aim of this work was to determine the influence of the addition of different amounts of waste coffee grounds (WCG) on the microstructure, morphology, density, porosity and shrinkage of alumina ceramics. Next research will be focused on the investigation of mechanical and tribological properties of mentioned samples.
- In Figure 5B, error bars should be added.
Response - Thank you for the observation. We have added error bars in Figure 5B in the revised manuscript.
- XRD pattern of Al2O3 ceramics should be added.
Response - Thank you for this valuable comment. We have added in Figure 4 XRD pattern of all alumina samples (without and with the addition of different amount of waste coffee grounds), raw WCG and thermally treated WCG at 1600 °C. We also discussed the obtained results.
- In Figure 6, the grain size distribution of Al2O3 ceramics with different waste coffee grounds should be provided.
Response – The crystallite size of all sintered alumina samples was measured from XRD patterns (Figure 4) using Scherrer's method. Obtained results are presented in Table 3. Alumina crystallite sizes (Table 3) increased with the addition of 1 wt. % of WCG. By further increasing the amount of WCG (5, 10 and 15 wt. %) the alumina crystallite size is not in linear correlation.
- In line 278-279, It was found that increasing the waste coffee grounds amount reduces the crystal grains of sintered Al2O3 ceramics and increases porosity. Please explain why?
Response - The burning of waste coffee grounds (WCG) has a major effect on the porous structure of the samples, similar results observed for clay with 5, 10, 15 and 20 wt. % of coffee waste [ref. 26].
- In Figure 5C, the shrinkage of the obtained foam is almost not affected by the amount of the waste coffee grounds. This is not consistent with the conclusion (Line 285-286).
Response – We agree with the reviewer. The results of the shrinkage of the Al2O3 ceramic samples (Figure 5 C) show a small change by addition of WCG from 0 wt. % to 20 wt. %.
- References should be formatted in accordance with ‘Instructions for Authors’.
Response - Thank you for the observation. We corrected references according to the ‘Instructions for Authors’.
- In addition, the English expression needs to be further improved by a professional scientific writer.
Response – Corrected.

Reviewer 2 Report
Porous alumina ceramics can be applied in manifold applications such as filters, membranes, catalyst supports, refractory and construction materials, and thermal insulators. The work focuses on the use of waste coffee grounds for the sacrificial fugitive method for obtaining porous alumina ceramics. With this upcycling approach, the work fits the aim and scope of the journal. However, the following points need to be addressed before publication:
1. Section 3.2 and Figure 3: the unit of the heating rate is K/min.
2. Fig. 4 should be implemented in section 2.1 as it shows the phase composition of the alumina raw material.
3. Please provide the chemical composition of the waste coffee ground. It is known, that coffee ground contains nitrogen, phosphor, and - most important – potassium. Alkalines are sintering agents for alumina, which will explain the increasing shrinkage with an increasing amount of WCG (one would expect no difference or an expansion due to degassing effects. Please add such a discussion to your manuscript. And please also provide the XRD of the samples after sintering.
4. Please provide a reference of alumina with e.g. 15 wt.% of a state-of-the-art pore-forming agent, so that the reader could comprehend your results. Especially the porosity and pore size distribution of the materials after sintering would be very interesting.
Author Response
Dear Editor and Reviewer,
Thank you for the review. We have done our best to correct the article and fulfill all the requirements and questions this time. We are resubmitting the corrected manuscript as suggested by the reviewers. The added or changed text was highlighted in “yellow”.
Our responses are as follows:
Reviewer #2:
Porous alumina ceramics can be applied in manifold applications such as filters, membranes, catalyst supports, refractory and construction materials, and thermal insulators. The work focuses on the use of waste coffee grounds for the sacrificial fugitive method for obtaining porous alumina ceramics. With this upcycling approach, the work fits the aim and scope of the journal. However, the following points need to be addressed before publication:
- Section 3.2 and Figure 3: the unit of the heating rate is K/min.
Response – Generally, for the heating rate the used unit is °C/min (we have to use this unit for programming of the sintering regime of our electrical kiln).
- Fig. 4 should be implemented in section 2.1 as it shows the phase composition of the alumina raw material.
Response – We have provided XRD patterns of Al2O3 raw powder, alumina ceramic sample with different amounts of WCG (0 wt. %, 1 wt. %, 5 wt., 10 wt. and 15 wt. %), WCG raw, WCG thermally treated at 1600 °C in section Results and discussion. Also, we discussed the obtained results (phase composition of all samples) in the revised manuscript.
- Please provide the chemical composition of the waste coffee ground. It is known, that coffee ground contains nitrogen, phosphor, and - most important – potassium. Alkalines are sintering agents for alumina, which will explain the increasing shrinkage with an increasing amount of WCG (one would expect no difference or an expansion due to degassing effects. Please add such a discussion to your manuscript. And please also provide the XRD of the samples after sintering.
Response – Thank you for your comments. We have provided XRD patterns of Al2O3 raw powder, alumina ceramic sample with different amounts of WCG (0 wt. %, 1 wt. %, 5 wt., 10 wt. and 15 wt. %), WCG raw, WCG thermal treated at 1600 °C with discussion of the obtained results (phase composition of all samples. Also, we performed and discussed EDS results (EDS spectra of Al, O, K and Mg of Al2O3 powder raw, alumina ceramics sample with different amounts of WCG (0 wt. %, 1 wt. %, 5 wt., 10 wt. and 15 wt. %), WCG raw, WCG thermally treated at 1600 °C).
- Please provide a reference of alumina with e.g. 15 wt.% of a state-of-the-art pore-forming agent, so that the reader could comprehend your results. Especially the porosity and pore size distribution of the materials after sintering would be very interesting.
Response - Thank you for your advice. However, it is quite difficult to find a reference to compare with our research since there are various methods for the preparation of alumina ceramics, as well as a great number of different pore-forming agents. The wide scope of different possible usage of the obtained porous ceramics makes it even harder to compare. The results obtained by other researchers give us only a small insight into the preparation process of porous alumina ceramics using waste as a pore-forming agent. That is why we intend to further broaden our research as well as characterization.

Reviewer 3 Report
The manuscript (sustainability-1882274) titled “Reusing of waste coffee grounds for preparation of porous alumina ceramics” investigated the effect of waste coffee ground contents on porous alumina ceramic properties, which is similar with the previous study of “Julian Alzukaimi, Rafi Jabrah. The preparation and characterization of porous alumina ceramics using an eco-friendly pore-forming agent. International Journal of Applied Ceramic Technology. 16(2); 820-831”. Thus, my decision is a major revision. My recommendations for this manuscript include:
1. The novelty of the study needs to be considered and provided in the introduction section.
2. How did the use of waste coffee ground to prepare porous alumina ceramic in this study differ from the existing studies? This should be mentioned in the introduction section.
3. Abbreviations of “Waste coffee ground” need to be defined at first mention and used consistently thereafter. Please check the whole manuscript.
4. References need to be provided more in Section 3 (Results and discussion section).
5. The authors need to discuss your study results with the other related studies, and provide the theory reasons to support your study results.
6. Why did the shear rate increase with an increase in WCG amount? This should be provided in section 3.1.
7. The references need to be provided in section 3.2.
8. Why did the shrinkage decrease in dimensions after sintering?
9. Why did the density, relative density, total porosity, and shrinkage of the sintered alumina ceramics change with WCG amount?
10. The equations (2) to (5) need to be moved from section 3 (Results and discussion section) to section 2 (Materials and method section).
11. In the conclusion section, suggestions or recommendations need to be provided.
Author Response
Dear Editor and Reviewer,
Thank you for the review. We have done our best to correct the article and fulfill all the requirements and questions this time. We are resubmitting the corrected manuscript as suggested by the reviewers. The added or changed text was highlighted in “yellow”.
Our responses are as follows:
Reviewer #3:
The manuscript (sustainability-1882274) titled “Reusing of waste coffee grounds for preparation of porous alumina ceramics” investigated the effect of waste coffee ground contents on porous alumina ceramic properties, which is similar with the previous study of “Julian Alzukaimi, Rafi Jabrah. The preparation and characterization of porous alumina ceramics using an eco-friendly pore-forming agent. International Journal of Applied Ceramic Technology. 16(2); 820-831”. Thus, my decision is a major revision. My recommendations for this manuscript include:
- The novelty of the study needs to be considered and provided in the introduction section.
Response – We have added in introduction.
- How did the use of waste coffee ground to prepare porous alumina ceramic in this study differ from the existing studies? This should be mentioned in the introduction section.
Response – Thank you. We have added in introduction section.
- Abbreviations of “Waste coffee ground” need to be defined at first mention and used consistently thereafter. Please check the whole manuscript.
Response – Thank you for comments. We have added in abstract.
- References need to be provided more in Section 3 (Results and discussion section).
Response – Thank you for your advice. We have added.
- The authors need to discuss your study results with the other related studies, and provide the theory reasons to support your study results.
Response – Thank you for your advice. The present paper aimed to provide further knowledge on the influence of different waste coffee grounds amounts used to produce alumina ceramics with different porosity. In addition, valuable information of other alumina ceramics samples properties (density, porosity, shrinkage, phase composition, crystallite size, morphology as well as EDS analysis) are provided.
- Why did the shear rate increase with an increase in WCG amount? This should be provided in section 3.1.
Response – From our experience, we know that if we change powder mixture composition, the shear rate will also change.
- The references need to be provided in section 3.2.
Response – Thank you for your comments. We have added.
- Why did the shrinkage decrease in dimensions after sintering?
Response - The results of the shrinkage of the Al2O3 ceramic samples (Figure 5 C) show a small change by addition of WCG from 0 wt. % to 20 wt. %. We corrected in manuscript.
- Why did the density, relative density, total porosity, and shrinkage of the sintered alumina ceramics change with WCG amount?
Response - The burning of waste coffee grounds (WCG) has a major effect on the porous structure of the samples, as well as for density.
- The equations (2) to (5) need to be moved from section 3 (Results and discussion section) to section 2 (Materials and method section).
Response -Thank you for your observation. We have corrected.
- In the conclusion section, suggestions or recommendations need to be provided.
Response –Thank you for your advice. We have added in conclusion: “Mechanical and tribological properties of alumina ceramics with addition different amount of WCG will be investigated in future research work.”

Round 2
Reviewer 1 Report
The crystallite size of all sintered alumina samples was measured by using Scherrer's method, when the values are higher than 100 nm, it is meaningless.
Author Response
Response to Reviewer’s Comments
Manuscript reference number: sustainability-1882274
The paper titled “Reusing of waste coffee grounds for preparation of porous alumina ceramics”
Dear Editor and Reviewer,
Thank you for the review. We have done our best to correct the article and fulfill all the requirements and questions this time. We are resubmitting the corrected manuscript as suggested by the reviewers. The added or changed text was highlighted in “green”.
Our responses are as follows:
Reviewer #2:
The crystallite size of all sintered alumina samples was measured by using Scherrer's method, when the values are higher than 100 nm, it is meaningless.
Response – Thank you for your valuable comment. The text was modified to enhance clarity. Bit generally, for crystallites up to 100 nm in size, the application of the Scherrer equation yields very reliable and precise estimation of the crystallite sizes. That does not limit the broader range of applicability, where in the range 20–200 nm the precision is still high, and for trends without enforcing precision the range can be broadened even up to 300 nm. Anyway, Scherrer equation suffers from a set of other assumptions, so saying that up to 100 nm the values are correct and above are not just inappropriate. In terms of trend, that is pretty much, what we wanted to show, similarly to numerous authors in numerous publications nowadays. We really even do not insist on the absolute value of the determined crystallite sizes. [New reference 26: P. Scherrer; Bestimmung der Gröss und der InnerenStruktur vonKolloidteilchen Mittels Rontgenstrahlen Nachrichten von der Gesellschaft der Wissenschaften, Göttingen,” Mathematisch-Physikalische Klasse, Vol. 2, 1918, pp. 98-100].
